# VMP1 Regulated by chi-miR-124a Effects Goat Myoblast Proliferation, Autophagy, and Apoptosis through the PI3K/ULK1/mTOR Signaling Pathway

**DOI:** 10.3390/cells11142227

**Published:** 2022-07-18

**Authors:** Yufang Liu, Zuyang Zhou, Kunyu Li, Peng Wang, Yulin Chen, Shoulong Deng, Wenting Li, Kun Yu, Kejun Wang

**Affiliations:** 1College of Animal Sciences and Technology, Henan Agricultural University, Zhengzhou 450046, China; aigaiy@126.com (Y.L.); liwenting_5959@126.com (W.L.); 2College of Life Sciences and Food Engineering, Hebei University of Engineering, Handan 056021, China; zuyangzhou163@163.com (Z.Z.); kunyuli3370@163.com (K.L.); wp05223414@163.com (P.W.); 15139671127@163.com (Y.C.); 3Institute of Laboratory Animal Sciences, Chinese Academy of Medical Sciences and Peking Union Medical College, Beijing 100193, China; popo84350746@163.com; 4College of Animal Science and Technology, China Agricultural University, Beijing 100193, China

**Keywords:** goat, longissimus dorsi tissue, apoptosis, autophagy, VMP1, miR-124a

## Abstract

The production of goat meat is determined by the growth speed of muscle fibers, and the autophagy and apoptosis of myoblast cells is a crucial process in the growth of muscle fibers. The rapid growth of muscle fibers occurs from one month old to nine months old in goats; however, the mechanisms of myoblast cells’ autophagy and apoptosis in this process are still unknown. To identify candidate genes and signaling pathway mechanisms involved in myoblast apoptosis and autophagy, we compared the expression characteristics of longissimus dorsi tissues from Wu’an goats—a native goat breed of China—at 1 month old (mon1 group) and 9 months old (mon9 group). Herein, a total of 182 differentially expressed mRNAs (DEGs) in the mon1 vs. mon9 comparison, along with the KEGG enrichments, showed that the PI3K-Akt pathway associated with autophagy and apoptosis was significantly enriched. Among these DEGs, expression of vacuole membrane protein 1 (VMP1)—a key gene for the PI3K-Akt pathway—was significantly upregulated in the older goats relative to the 1-month-old goats. We demonstrated that VMP1 promotes the proliferation and autophagy of myoblasts, and inhibits their apoptosis. The integration analysis of miRNA–mRNA showed that miR-124a was a regulator of VMP1 in muscle tissue, and overexpression and inhibition of miR-124a suppressed the proliferation and autophagy of myoblasts. The PI3K/Akt/mTOR pathway was an important pathway for cell autophagy. Additionally, the activator of the PI3K/Akt/mTOR pathway, the expression of VMP1, and ULK1 were higher than the negative control, and the expression of mTOR was depressed. The expression of VMP1, ULK1, and mTOR was the opposite when the inhibitor was added to the myoblasts. These results show that the PI3K/Akt/mTOR pathway promoted the expression of VMP1 and ULK1. By using adenovirus-mediated apoptosis and proliferation assays, we found that that miR-124a inhibits myoblast proliferation and autophagy, and promotes their apoptosis by targeting VMP1. In conclusion, our results indicated that VMP1 was highly expressed in the LD muscle tissues of nine-month-old goats, and that it was regulated by miR-124a to inhibit myoblast cells’ apoptosis through the PI3K/Akt/mTOR pathway, and to promote proliferation and autophagy. These findings contribute to the understanding of the molecular mechanisms involved in myoblast proliferation, autophagy, and apoptosis.

## 1. Introduction

Skeletal muscle’s metabolic plasticity, growth, and atrophy have been widely investigated in past decades—particularly the effects that systemic inflammation has on these processes [1]. The domestic goat (*Capra hircus*) is an important agricultural animal; goat meat has a higher protein content, but lower fat and cholesterol content, than beef or pork [2]. An improved understanding of the molecular mechanisms that regulate skeletal muscle growth and development may result in an opportunity to enhance meat production [3]. A larger diameter of skeletal muscle fiber corresponds to a faster growth rate; in goats, the diameter of muscle fibers increases between 1 and 9 months of age, and then remains constant [4]. Skeletal muscle groups originate from different regions of the developing embryo, and have characteristic morphological specializations [5,6,7]. During the development and regeneration of skeletal muscle, myogenic precursors undergo several cellular and molecular changes, differentiating into multinucleated myofibers [8]. When myoblasts are stimulated by regulatory factors, they begin to migrate to the target tissue area and proliferate extensively. Under the actions of myoblast regulatory factors and muscle regulatory factors (MRFs), the cells differentiate and fuse to form multinucleated myotubes and, eventually, muscle tissue [9]. The mechanisms of skeletal muscle development and myoblast proliferation, however, remain unclear.

In recent years, RNA-Seq has been widely used to analyze the genetic mechanisms underlying skeletal muscle growth and development in pigs [10,11], cattle [12,13], and sheep [14,15]. Between fetal and juvenile Huanghuai goats, Wang et al. (2015) identified 6432 differentially expressed genes in longissimus thoracis muscle tissues; many of these genes were found to be involved in fetal myogenesis, proliferation, and the differentiation of muscle cells [16]. In the longissimus dorsi muscle tissues of Jianzhou big-eared goats, Lin et al., 2017 reported 111 differentially expressed genes between kids (2-month-old), young (9-month-old), and adult (24-month-old) goats; these DEGs were related to muscle development and lipid metabolism [17]. Some important functional genes and signaling pathways related to meat yield and quality have been found. For example, previous studies have revealed the crucial roles of myostatin (MSTN), MRF family members, and insulin-like growth factor (IGF) family members in skeletal muscle growth and development in goats [18,19,20]. The autophagy signaling pathway is especially important for energy generation/consumption and macromolecule turnover processes in skeletal muscles [21]. In mammalian cells, the sequential association of at least a subset of the ATG proteins, referred to as the core molecular machinery [22], leads to autophagosome formation. VMP1 belongs to this family of essential ATG proteins. In recent years, the biological function of VMP1 has been increasingly elucidated, and when VMP1 is located on the ER membrane, it has been shown to promote the formation of autophagosomes by regulating the interaction between the endoplasmic reticulum (ER) and isolated membranes [23,24,25]. Conversely, autophagy is completely blocked in the absence of VMP1 [26].

In this study, we used RNA sequencing to determine the differentially expressed mRNAs in longissimus dorsi (LD) tissues of Wu’an goats at different stages of growth: 1-month-old goats (average fiber diameter, 9.12 ± 0.13 μm) and 9-month-old goats (average fiber diameter, 28.9 ± 1.42 μm). We further characterized the autophagy-associated gene VMP1 as a target of miR-124a and inhibitor of the rapamycin complex 1 (mTOR) signaling pathway, thereby regulating myoblast proliferation, autophagy, and apoptosis. Identifying the regulatory function of VMP1 and its role in signaling pathways associated with the development of skeletal muscle and myoblast proliferation should generate information that would be useful to improve meat yield.

## 2. Materials and Methods

### 2.1. Ethics Statement

All experiments involving animals were approved by the Institutional Animal Care and Use Committee (IACUC) of Henan Agricultural University (Permit Number: 17-0118).

### 2.2. Animals and Sample Collection

Ten female Wu’an black goats—three aged 1 month (mon1) and three aged 9 months (mon9) were selected for the RNA-Seq. These goats were bred at a goat breeding farm in the city of Wu’an, Hebei Province, China. All goats were raised under the same conditions to minimize differences arising from external factors. Immediately after slaughter, longissimus dorsi tissue was taken and placed on dry ice until storage at −80 °C. A selection of tissues was immediately put into a 4% paraformaldehyde fixation solution for use in preparing frozen sections.

### 2.3. Hematoxylin–Eosin Staining

Histology was performed using conventional methods on longissimus dorsi (LD) tissues that had been preserved in 4% paraformaldehyde for 72 h. Hematoxylin–eosin staining was performed according to the protocol of Guardiola et al. [27]. Samples were observed using a fluorescence microscope (Olympus, Tokyo, Japan). ImageJ software was used to analyze the diameter of muscle fibers from each group.

### 2.4. RNA Extraction, Library Construction, and Differential Expression Analysis

Total RNA from the longissimus dorsi tissue was isolated using TRIzol reagent (Invitrogen, Carlsbad, CA, USA). The quality, concentration, and integrity of the RNA were assessed using a NanoDrop photometer and an Agilent 2100 bioanalyzer. The values for 260/280 ranged from 1.8 to 2.0. Six cDNA libraries were constructed and subjected to Illumina HiSeq 2500 high-throughput sequencing. The RIN values ranged from 7.8 to 8.6, in line with the requirements of the library construction. Construction and sequencing of the mRNA libraries were as described by Jia et al. [28]. Prior to differential gene expression analysis, read counts for each sequenced library were normalized using the edgeR R package, as described by Law CW et al., 2016 [29]. Differentially expressed genes were identified by comparing normalized mon1 and mon9 reads using the DEGseq2 (2010) R package as described by Love MI et al., 2014 [30]. *p*-Values were adjusted using the q-value [31]. Genes exhibiting a *q*-value < 0.005 and |log2(fold change)| > 1 were classified as differentially expressed genes (DEGs). Upregulated genes were defined as those whose transcripts were more abundant in the mon9 libraries.

### 2.5. Gene Ontology (GO) and Kyoto Encyclopedia of Genes and Genomes (KEGG) Pathway Analyses

To associate DEGs with biological and metabolic pathways, GO (http://geneontology.org, accessed on 5 December 2021) and KEGG (www.genome.jp/kegg, accessed on 5 December 2021) analyses were performed using DAVID Bioinformatics Resources v6.7 (http://david.abcc.ncifcrf.gov/, accessed on 5 December 2021). We used KOBAS to test for statistically significant enrichment of gene candidates in the KEGG pathways [32].

### 2.6. Reverse-Transcription (RT)-qPCR Verification

RT-qPCR was used to verify the expression levels of the differentially expressed genes and miRNAs. Approximately 0.1 μg of RNA per sample was reverse transcribed into cDNA using RT reagents (Takara, Dalian, China). GAPDH and U6 were used as endogenous controls for normalizing the expression of genes and miRNAs. All experiments were performed with five biological replicates, and each sample was tested in triplicate. RT-qPCR was performed on a LightCycler 480II (Roche, Basel, Sweden) using SYBR Premix Ex Taq II. The cycling conditions were pre-denaturation at 95 °C for 5 s, then 40 cycles of 95 °C for 5 s, and 60 °C for 30 s. Melting curve analysis was performed, and the relative expression levels were determined using the 2 ^ΔΔCt^ method [33]. The *p*-value calculation was performed by *t*-test, and *p* < 0.05 was used to indicate significant differences. The primers were designed using Primer 5 (listed in Appendix A).

### 2.7. miRNA Prediction for VMP1 Targeting

Integrated analysis of our previous miRNA-Seq sequencing data (PRJNA831288) revealed that miR-124a was able to directly target VMP1. VMP1 3′-UTR sequences were obtained from NCBI (XM_018064144.1). Interactions between VMP1 and miRNA were predicted vs. the mature goat miRNA sequence using TargetScan (http://www.targetscan.org/vert_71/, accessed on 7 February 2022) and miRanda (https://www.miranda-ng.org/en/, accessed on 7 February 2022). Mature miRNA sequences from other species were obtained from miRBase (http://www.mirbase.org, accessed on 10 February 2022).

### 2.8. Cell Culture

Following previously described methods, primary goat myoblasts were isolated from the longissimus dorsi of goats (Fauconneau and Paboeuf, 2000) [34]. The isolated cells were seeded into 6 mm plates and cultured in a complete medium (DMEM/F12 [1:1], 20% FBS, and 1% penicillin/streptomycin) (Gibco, Melbourne, Australia), as described by Duran et al. [35]. When cells reached greater than 90% confluence, they were transferred to 10 cm plates for the experiment.

### 2.9. Plasmid Construction and Transfection

The coding region of VMP1 was cloned into a pcDNA3.1 expression vector, and the recombinant vectors were named pcDNA3.1-VMP1 and pcDNA3.1-NC. The 3′-UTR of wild-type and mutant VMP1, containing the predicted target sites, were each cloned into a pmiR-GLO vector between Xho I and Not I; the recombinant vectors were named pmiR-GLO-VMP1-WT and pmiR-GLO-VMP1-MUT. The siRNA-VMP1, siRNA-NC, and the miR-124a mimic and inhibitor were synthesized by Genewiz (Suzhou, China).

### 2.10. Dual-Luciferase Reporter Assay

293T cells, a human renal epithelial cell line, were cultured in DMEM and 10% FBS and used to validate the miRNA target. Cells were seeded into 24-well plates, then co-transfected with 200 ng of mRNA-3′-UTR-WT or mRNA-3′-UTR-MUT together with 10 µL of miRNA mimic or mimic-NC using Lipofectamine 2000 (Invitrogen, USA). At 48 h post-transfection, luciferase activity was measured using the Dual Luciferase Reporter Assay System (Promega, WI, USA). Samples were assayed in triplicate.

### 2.11. Western Blot Analysis

Protein was extracted from goat LD tissues and primary myoblast cells using a Total Histone Extraction Kit (Beyotime Biotech Co., Ltd., Shanghai, China); protein concentrations were determined using a BCA protein assay kit (Beyotime). The 30 µg of protein per sample was resolved using 12% SDS–PAGE and then transferred onto PVD membranes activated with methanol. After transfer, membranes were blocked with blocking buffer (Beyotime) for 6 h at 4 °C, and then incubated with primary antibodies overnight at 4 °C. The primary antibodies were anti-ULK1 (1:100) (Santa Cruz Biotechnology, New York, NY, USA), anti-VMP1 (1:500), anti-beclin-1 (1:1000), anti-caspase-3 (1:1000), anti-mTOR mammalian (1:1000), and anti-GAPDH (1:1000) (all from Cell Signaling Technology, New York, NY, USA). After incubation, the membranes were rinsed with wash buffer (Beyotime) and incubated for 2 h at 4 °C with secondary antibodies. The secondary Ab was HRP-conjugated goat anti-rabbit IgG (H + L) (Proteintech, Wuhan, China), used at dilutions recommended by Beyotime.

### 2.12. Cell Proliferation Assay

The 1 × 10^6^ myoblast cells were inoculated into wells of 96-well plates and then incubated for 2–4 h at 37 °C. Cell proliferation was measured using a Cell Counting Kit-8 (CCK-8) (Solarbio, Beijing, China) according to the manufacturer’s protocol. Absorbance at 450 nm was measured at 24, 48, and 72 h after the addition of the CCK-8 solution.

### 2.13. Ethynyldeoxyuridine Incorporation Assay

Myoblast proliferation was quantified using an ethynyldeoxyuridine (EdU) kit according to the manufacturer’s protocol (Beyotime). Cells were seeded into 96-well plates as described above. Them, 100 µL of 50 µM EdU was aliquoted to each well, and the cells were incubated for an additional 3 h. The cells were then washed with PBS and fixed with 4% paraformaldehyde for 30 min. To neutralize excess aldehyde groups, 50 µL of 2 mg/mL of glycine was aliquoted per well and incubated with cells for 15 min. Subsequently, 100 µL of 0.5% Triton X-100 in PBS was aliquoted per well and incubated with cells for 15 min. After rinsing, 100 µL of Apollo reagent was added, and the cells were incubated in the dark for 30 min at room temperature. Cells were washed with PBS and then their nuclei were stained with Hoechst 33,342 reaction solution for 30 min in the dark. EdU-stained cells were visualized and quantified using a fluorescence microscope. Three fields were randomly selected for quantification and statistical analysis.

### 2.14. Cell Apoptosis Analysis

Myoblasts were washed twice with PBS, and the concentration was adjusted to 10^6^ cells/mL. The 100 µL of binding buffer (MeilunBio, Dalian, China) was then added to the cell suspension along with 5 µL of annexin V–FITC (MeilunBio), cells were then incubated for 10 min at room temperature. Cells were then incubated with 7.5 µL of propidium iodide (PI, MeilunBio) for 15 min at room temperature in the dark. Apoptosis was analyzed by fluorescence microscopy, as well as by flow cytometry using FlowJo software (v7.6.1, Ashland, OR, USA).

### 2.15. PI3K/ULK1/mTOR Pathway Analysis

3-Methyladenine (3-MA, S1039)—a PI3K inhibitor—and rapamycin (AY-22989, S2767)—a PI3K activator—were used to investigate the relationship between PI3K and VMP1. Myoblast cells were incubated with 3-MA or AY-22989 for 3 h as described by Zhang et al. [36]. A CCK-8 assay was used to determine the optimal concentrations of 3-MA and AY-22989 for use in the experiments. The 3-MA and AY-22989 were purchased from Selleck Chemicals (Houston, TX, USA).

### 2.16. Confocal Microscopy

The extent of the autophagic flux was evaluated using an adenovirus-harboring tandem fluorescent mRFP-GFP-LC3 (Hanbio, Shanghai, China). Myoblasts were grown on glass slides in six-well plates and transfected with 4 µg of pcDNA3.1-VMP1, pcDNA3.1-NC, si-VMP1, si-NC, miR-124a mimic, mimic NC, miR-124a inhibitor, or inhibitor NC. After incubation for 1 h, 500 MOI (multiplicity of infection) of adenovirus was aliquoted to cells and incubated for 6–8 h. The culture medium was then changed, and the cells continued incubating for a total of 48 h. Subsequently, the cells were washed with cold PBS and fixed with 4% paraformaldehyde for 30 min. The cells were washed three times with PBS and then observed using a confocal microscope (Olympus, Melville, NY, USA).

### 2.17. Statistical Analysis

Data were subjected to statistical analyses using SPSS 20 statistical software, and the mean of three replicates was evaluated and displayed as the mean ± standard error (SE). The histograms were completed using Excel and GraphPad Prism. Significance was determined using Duncan’s multiple range tests, and is presented as * *p* < 0.05 and ** *p* < 0.01.

## 3. Results

Histological observations revealed distinct differences in the morphological characteristics of longissimus dorsi (LD) from the mon9 and mon1 groups (Figure 1). The average diameter of mon9 muscle fibers (34.68 ± 2.18 μm) was significantly greater than the diameter of mon1 fibers (18.48 ± 1.29 μm).

### 3.1. RNA-Seq Analysis

Six mRNA libraries were constructed (3 each for the mon1 and mon9 groups). Bulk statistics for reading counts and mapping rates are presented in Appendix A. A total of 46,929 known transcripts and 12,434 novel transcripts were identified (Appendix A) and used in subsequent analyses. After a series of filtering steps, 182 genes were classified as differentially expressed (Appendix A). Among them, 37 were upregulated and 145 downregulated (Figure 2A). The DEG expression profiles are summarized as a heat map in Figure 2B and Appendix A. The GO analysis showed that the DEGs were involved primarily in cell growth, cell migration, material transport, cell adhesion regulation, cell component movement regulation, and material synthesis (Figure 2C). However, the KEGG pathways containing the most clustered DEGs were the PI3K-AKT signaling (chx04151) and p53 signaling pathways (chx04115) (Figure 2D).

We selected 10 DEGs and independently verified their expression using RT-qPCR, including five downregulated genes (ANKRD1, RASD1, XIRP1, KLHL40, and TWF2) and five upregulated genes (ADAMTS2, NREP, COL1A1, FGF18, and HBBC). A comparison of the relative gene expression levels determined by RT-qPCR and RNA-Seq showed that the two methods yield consistent results (Figure 2E).

### 3.2. Function of VMP1 in Cell Proliferation and Apoptosis of Goat Myoblasts

The growth and development of the longissimus dorsi are mainly determined by cell proliferation, apoptosis, and differentiation [37]. VMP1 was significantly upregulated in mon9 relative to mon1 goats, and was enriched in the PI3K-mTOR pathway, which functions in cell proliferation and apoptosis processes. To determine the role of VMP1 in goat myoblast cell proliferation and apoptosis, VMP1 overexpression and knock-down models were constructed (Figure 3A,B). VMP1 overexpression induced an increase in mRNA levels of the proliferation marker genes CDK4, cyclin D1, and cyclin D2 (Figure 3E). A CCK-8 assay revealed that cell proliferation was induced 6 h after transfection in the overexpression model (Figure 3C). Additionally, increased quantities of EdU-positive cells were observed when VMP1 was overexpressed (Figure 3G,H). These results demonstrate that overexpression of VMP1 promotes the proliferation of goat myoblasts. In contrast, we observed the opposite pattern in the knockdown experiment using si-VMP1. The results show that proliferation marker genes were downregulated (Figure 3F), cell proliferation was reduced 24 h post-transfection (Figure 3D), and EdU-positive cell numbers decreased (Figure 3G,I). Overall, these results demonstrate that knockdown of VMP1 suppresses goat myoblast cell proliferation.

We then investigated the effects of VMP1 on goat cell apoptosis. Apoptosis was inhibited by transfection with pcDNA3.1-VMP1. VMP1 overexpression was associated with a decrease in mRNA and protein levels of caspase-9, caspase-3, and beclin-1 (Figure 4A,C,D). The siRNA-VMP1 knockdown showed the opposite trend, with increased expression of caspase-9, caspase-3, and beclin-1 (Figure 4B–D). Flow cytometry revealed that transfection with pcDNA3.1-NC increased the numbers of apoptotic cells, compared with transfection with pcDNA3.1-VMP1 (Figure 4E,F). In contrast, apoptotic cell numbers decreased slightly after siRNA-NC transfection compared with siRNA-VMP1 transfection (Figure 4G,H). Together, these results indicate that VMP1 inhibits apoptosis in goat myoblast cells.

### 3.3. miR-124a Targets VMP1

Integrated analysis of our previous miRNA-Seq sequencing data (PRJNA831288) revealed that miR-124a was able to directly target VMP1. Consistent results were predicted by both TargetScan and miRanda software. The RT-qPCR results showed that the expression of miR-124a was significantly higher in LD tissues in the mon1 group than in the mon9 group (Figure 5A). The expression level of the VMP1 protein was significantly lower in the mon1 group than in the mon9 group, which was also consistent with their mRNA expression (Figure 5B). Dual-luciferase reporter assays were performed to determine whether there was a direct interaction between miR-124a and VMP1. Luciferase activity of the VMP1 wild-type plasmid with the miR-124a mimic was lower than that of the VMP1 mutant plasmid with the miR-124a mimic (Figure 5C), supporting the hypothesis that VMP1 is a direct target of miR-214a. VMP1 mRNA and protein expression in goat myoblasts were determined after transfection with the miR-124a mimic or miR-124a inhibitor. The results showed decreased mRNA (Figure 5D) and protein (Figure 5E,F) expression in the miR-124a mimic transfected cells. In the miR-124a inhibitor transfected cells, the expression levels of VMP1 mRNA and protein were significantly increased (Figure 5E–G). These data demonstrate that the VMP1 gene is a direct target of miR-124a in goats.

### 3.4. miR-124a Inhibits Proliferation of, and Induces Cell Apoptosis in, Goat Myoblasts

To investigate the function of miR-124a in goat myoblast proliferation, myoblast cells overexpressing the miR-124a mimic showed significantly elevated levels of miR-124a (Figure 6A), as well as a decrease in the mRNA levels of proliferation-related genes (i.e., CDK4, cyclin D1, and cyclin D2) (Figure 6E). CCK-8 assays indicated that the proliferative state of these myoblast cells was depressed between 24 and 48 h post-transfection (Figure 6C). Additionally, from EdU staining we saw that the quantity of EdU-positive cells 24 h post-transfection was reduced in the miR-124a-mimic-transfected cells (Figure 6G,K). Cells transfected with an miR-124a inhibitor showed significantly decreased levels of miR-124a mRNA (Figure 6B), and significantly elevated levels of CDK4, cyclin D1, and cyclin D2 mRNAs (Figure 6F). CCK-8 assays indicated that the proliferative state of these cells was elevated between 24 and 48 h post-transfection (Figure 6D). From EdU staining, we saw that the quantity of EdU-positive cells was increased in the miR-124a-inhibitor-transfected cells (Figure 6G,H). Overall, these results demonstrate that miR-124a inhibits goat myoblast proliferation.

We also investigated the effect of miR-124a on myoblast apoptosis, and found that in cells transfected with the miR-124a mimic, the mRNA and protein levels of caspase-9, caspase-3, and beclin-1 were all increased (Figure 7A,C,D), indicating that apoptosis was promoted in these cells. miR-124-inhibitor-transfected cells showed decreased levels of caspase-9, caspase-3, and beclin-1 mRNA and proteins (Figure 7B–D). Flow cytometry revealed an increase in the number of apoptotic cells in the miR-124a mimic group compared with the mimic NC group (Figure 7E,F), and a slight decrease in the number of apoptotic cells in the miR-124 inhibitor group (Figure 7G,H). These results indicate that miR-124a promotes apoptosis in goat myoblasts.

### 3.5. VMP1 Promotes Goat Myoblast Proliferation via the PI3K/Akt/mTOR Pathway

To investigate the relationship between VMP1 and the PI3K signaling pathway, myoblasts were treated with a PI3K activator (AY-22989) or inhibitor (3-MA) (Figure 8A,B, respectively). Cells treated with AY-22989 (50 μg/mL) had increased levels of VMP1 and ULK1 over the AY-22989-NC-treated cells, and showed decreased levels of mTOR. Cells treated with 3-MA (20 nM) had increased levels of mTOR over the 3-MA-NC-treated cells, and showed decreased levels of VMP1 and ULK1 (Figure 8C,D). These results reveal that both VMP1 and ULK1 are downstream regulators of the PI3K/Akt/mTOR signaling pathways, and might be related to cell autophagy.

### 3.6. VMP1 Promotes Myoblast Autophagy

Recent studies have demonstrated that VMP1 is involved in the regulation of autophagy [38,39]. Therefore, we investigated the effects of VMP1 overexpression and knockdown on goat myoblast autophagy. Myoblasts transfected with pcDNA3.1-VMP1 and infected with the mRFP-GFP-LC3-harboring adenovirus had more GFP and mRFP puncta than cells transfected with pcDNA3.1-NC (Figure 9A–C). Cells transfected with siRNA-VMP1 had fewer red puncta than cells transfected with siRNA-NC, reflecting a decreased level of autophagic flux (Figure 9A,D,E). In cells transfected with pcDNA3.1-VMP1 (vs. pcDNA3.1-NC), levels of mTOR were decreased, while levels of ULK1 were increased. In contrast, in cells in which VMP1 had been knocked down, mTOR levels were increased, and ULK1 levels decreased; siRNA-VMP1 inhibited the degradation of mTOR and the accumulation of ULK1 (Figure 9F,G, respectively), thereby suppressing autophagy. These results demonstrate that VMP1 promotes autophagy by activating the VMP1/ULK1 pathway. Based on these results, we suggest that VMP1 promotes autophagy in goat myoblast cells.

### 3.7. miR-124a Regulates Myoblast Autophagy by Targeting VMP1

Transfected myoblasts were infected with an adenovirus harboring tandem fluorescent mRFP-GFP-LC3 in order to evaluate the extent of autophagic flux, allowing us to distinguish between autophagosomes and autolysosomes. The autolysosomes contained only mRFP, while the autophagosomes contained both GFP and red mRFP signals, and the merged images reveal a yellow signal where there is co-localization. Cells transfected with the miR-124a mimic had significantly fewer GFP and mRFP puncta per cell than cells transfected with the mimic NC (Figure 10A–C). In the overlaid images, fewer red dots were observed, indicating decreased autolysosome synthesis. Cells transfected with the miR-124a inhibitor had significantly more red puncta than cells transfected with the inhibitor NC, indicating an increased level of autophagic flux (Figure 10A,D,E). These data suggest that miR-124a inhibits goat myoblast autophagy.

### 3.8. miR-124a Impedes the VMP1/ULK1/mTOR Pathway

Myoblasts transfected with the miR-124a mimic had decreased levels of VMP1 and ULK1 (the expression of VMP1 shown in Figure 5F,G), but increased levels of mTOR. In contrast, cells transfected the miR-124a inhibitor had increased levels of VMP1 and ULK1, and decreased levels of mTOR. Inhibition of miR-124a resulted in the accumulation and degradation of VMP1 and ULK1, respectively (Figure 10F,G), resulting in enhanced autophagy. These results show that miR-124a inhibits autophagy by impeding the VMP1/ULK1/mTOR pathway.

## 4. Discussion

The response of myofibers to cancer and the cachectic environment is considered central to understanding the regulation of muscle development and wasting [40]. Abnormal autophagy in muscles results in cellular alterations, such as mitochondrial damage, ER stress, impaired sarcomeric protein turnover, and cell death [41], collectively leading to the development of various skeletal muscle diseases. Autophagic removal of dysfunctional organelles appears to be an important suppressor of the apoptotic death-signaling program [42,43]. In this study, we screened 182 DEGs from the LD tissues of 1-month-old and 9-month-old Wu’an goats. GO analysis showed that the DEGs corresponded to functions and pathways related to energy metabolism, substance metabolism, and cell movement. KEGG analysis showed that the majority of DEGs were enriched in the PI3K-Akt signaling pathway and regulation of the actin cytoskeleton. PI3K-Akt is a key signaling pathway involved in regulating the cell cycle; it is closely related to the proliferation and differentiation of skeletal muscle, along with muscle hypertrophy [44,45]. The DEGs we found enriched in the mon9 group included the significant PI3K-Akt genes, along with VMP1, which participated in the processes of cell proliferation, apoptosis, and autophagy [46,47].

High expression of VMP1 has been frequently linked to cancer, and is correlated with increased levels of cell proliferation and autophagy [39,48,49,50]. The aim of this study was to investigate the molecular mechanisms that underlie the differences in skeletal muscle between young growing goats (1-month-old) and goats that have finished growing (9-months-old). RNA-Seq showed that most DEGs were expressed at low levels in the longissimus dorsi tissues of young goats. We speculated that VMP1 plays an important role in goat myoblast proliferation, apoptosis, and autophagy. Integration of the previous study of miRNA-Seq data with bioinformatics tools analysis revealed that miR-124a is an important regulatory element of VMP1. Therefore, a series of experiments were conducted, and the results showed that miR-124a impeded goat myoblast proliferation and autophagy, and promoted apoptosis, by targeting VMP1. VMP1 (vacuole membrane protein 1) is a transmembrane protein associated with the ER, Golgi apparatus, and intracellular vesicles [51]. During early development, both protein synthesis and protein breakdown are equally important, and VMP1 plays a part in the proteostasis mechanism and removal of damaged organelles. In VMP1-depleted Cos-7 cells, which exhibited a fragmented ER and disorganized Golgi bodies, a rapid accumulation of AprA—a protein secreted via the ER–Golgi transport pathway—was observed in controls overexpressing VMP1, suggesting that VMP1 may also be involved in protein secretion [52,53]. Autophagy is the major degradation pathway involved in the clearance of protein aggregates; overexpression of VMP1 induces the formation of autophagosomes in mammalian cells, which might lead to clearance of accumulated proteins involved in neurodegenerative disorders [54,55]. It is also highly involved in the processes of protein secretion, phagocytosis, osmoregulation, and cytokinesis, thereby mediating diverse cellular processes [56]. Previous studies demonstrated that VMP1 plays an important role in the autophagic process by regulating interactions between the ER and the autophagic isolation membrane [23]. VMP1 is also involved in cellular processes and important signaling pathways, such as the AMPK pathway [57], PI3K–Akt pathway [58], and the mTOR pathway [59]. Morishita et al. reported that overexpression of VMP1 increased autophagic flux and improved mitochondrial quality, whereas its suppression resulted in decreased cell autophagy [60]. Here, we showed that overexpression of miR-124a promoted myoblast apoptosis, which was characterized by increased apoptotic cell numbers and increased expression of caspase-9, caspase-3, and beclin-1. The results of CCK-8 and EdU indicated that myoblasts overexpressing an miR-124a mimic had depressed viability and proliferation, and the opposite was true after inhibition of miR-124a expression. Subsequently, bioinformatics analyses and dual-luciferase reporter gene assays demonstrated that miR-124a directly targets VMP1. Expression levels of VMP1 mRNA and protein were suppressed in cells overexpressing miR-124a. VMP1-silenced myoblasts were inhibited in their proliferation, and the number of apoptotic cells was increased.

Increased expression of cleaved caspase-3 and caspase-9 leads to inhibition of autophagy and enhanced apoptosis [61]. We found that miR-124a regulates myoblast autophagy by targeting VMP1. Wang et al. reported that miR-21 directly inhibits the translation of VMP1, and that the loss of miR-21 leads to higher expression of VMP1 and stimulates autophagy [50]. In this study, we confirmed that VMP1 is a target gene of miR-124a and regulates myoblast autophagy by targeting the ULK1/mTOR pathway. The activity of mTOR—an upstream regulator of ULK1—prevents the activation of ULK1, and disrupts the interaction between ULK1 and AMPK [62]. The overexpression of miR-124a attenuated autophagic flux and reduced the expression of VMP1 and ULK1, while increasing the levels of mTOR. In contrast, miR-124a inhibition enhanced autophagic flux in cultured myoblasts, and led to increased expression of VMP1 and ULK1, along with decreased expression of mTOR. Inhibition of miR-124a resulted in the accumulation of ULK1 and VMP1 and the degradation of mTOR, indicating that miR-124a inhibited myoblasts’ proliferation and autophagy by impeding the VMP1/ULK1/mTOR pathway.

## 5. Conclusions

We annotated 182 DEGs between 1-month-old and 9-month-old Wu’an goats, and identified VMP1—predominantly expressed in the LD tissues of the older goats—as being involved in myoblast proliferation and autophagy and, thus, muscle development. We determined that VMP1 expression in myoblasts is regulated by miR-124a, and that by suppressing VMP1 expression, miR-124a inhibits the PI3K/ULK1/mTOR pathway, thereby inhibiting myoblast apoptosis and autophagy (Figure 11). This work provides insight into the regulatory mechanisms underlying muscle development, which could from the basis for the development of new therapeutic strategies for muscle proliferation and atrophy diseases.

## Figures and Tables

**Figure 1 cells-11-02227-f001:**
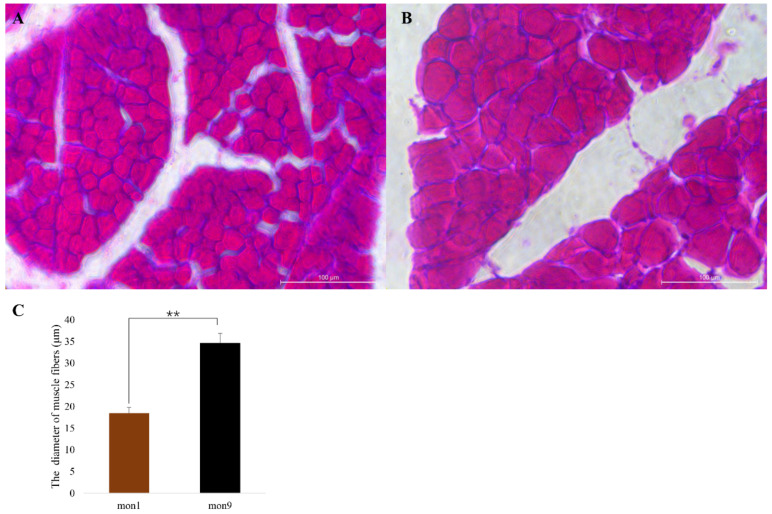
Diameter of longissimus dorsi fibers from Wu’an goats: (**A**) A representative transverse section of longissimus dorsi fibers from a 1-month-old goat. (**B**) A representative transverse section of longissimus dorsi fibers from a 9-month-old goat. (**C**) Muscle fiber diameter statistics for the mon1 and mon9 groups; three fibers per group were measured. Data are shown as the mean ± SE. ** *p* < 0.01.

**Figure 2 cells-11-02227-f002:**
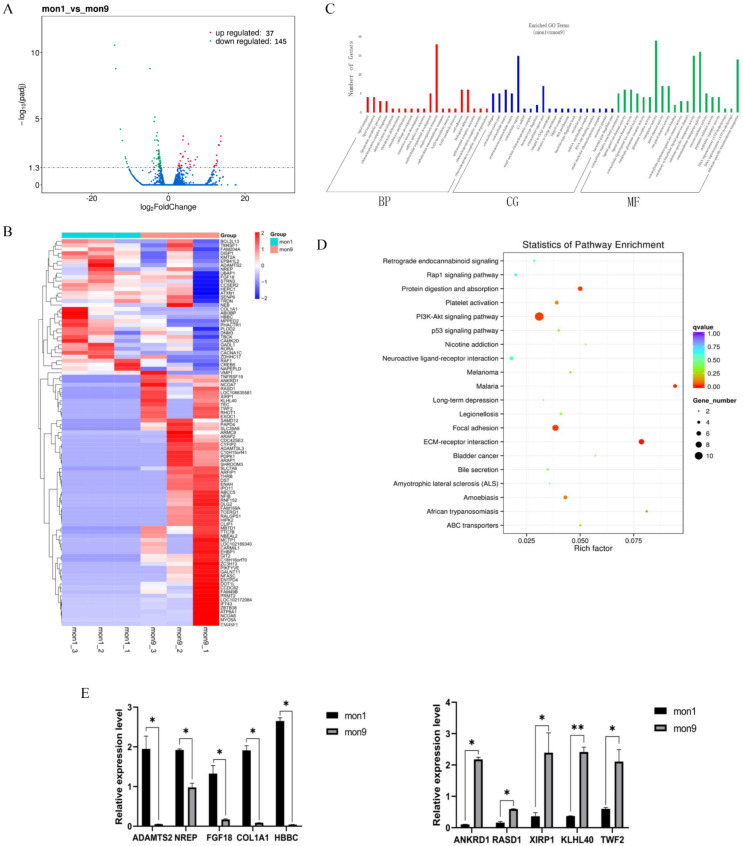
mRNA sequencing and analysis of differentially expressed genes (DEGs): (**A**,**B**) Volcano plot and heatmap of DEGs between mon1 and mon9. (**C**,**D**) Identified GO terms and KEGG pathways for the DEGs. (**E**) DEG verification by RT-qPCR and sequencing. * *p* < 0.05; ** *p* < 0.01.

**Figure 3 cells-11-02227-f003:**
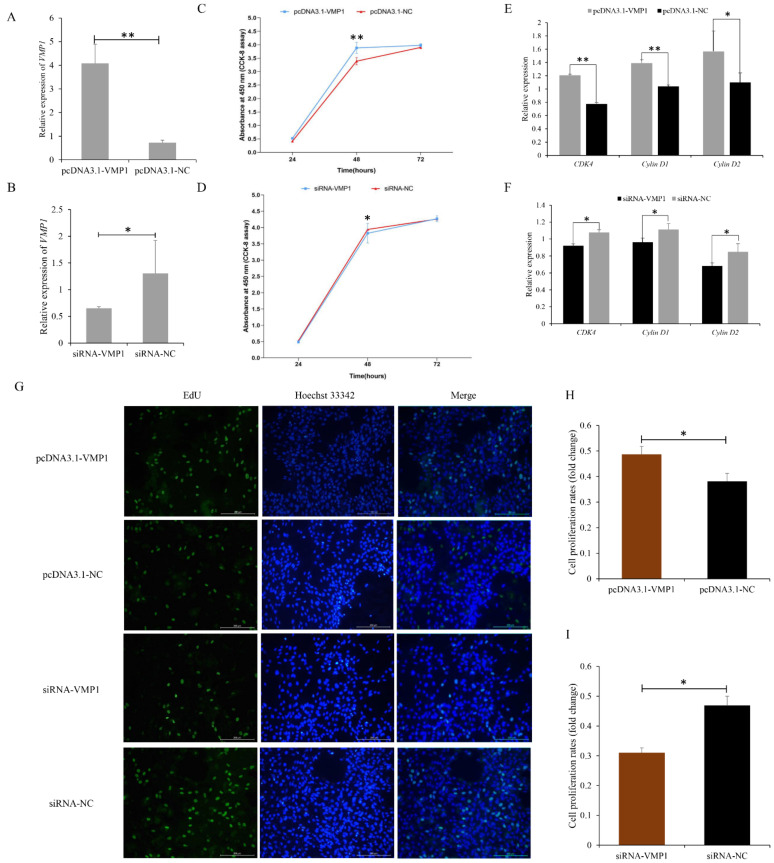
VMP1 regulates the proliferation of goat myoblasts: (**A**,**B**) RT-qPCR was used to determine VMP1 expression levels in myoblasts transfected with pcDNA3.1-VMP1 and siRNA-VMP1. (**C**) Proliferation of goat myoblasts overexpressing VMP1, measured using a Cell Counting Kit-8. (**D**) Proliferation of goat myoblasts transfected with siRNA-VMP1. (**E**) Expression of the cell-proliferation-related genes CDK4, cyclin D1, and cyclin D2 in cells overexpressing VMP1 was determined by RT-qPCR. (**F**) Expression of cell-proliferation-related genes in cells transfected with siRNA-VMP1. (**G**) EdU staining of myoblasts overexpressing VMP1, and in which VMP1 is silenced. (**H**,**I**) Fold change in the proliferation of myoblasts overexpressing VMP1, and in which VMP1 is silenced. Replicates = 3. Data are shown as the mean ± SE. * *p* < 0.05 and ** *p* < 0.01. NC, negative control.

**Figure 4 cells-11-02227-f004:**
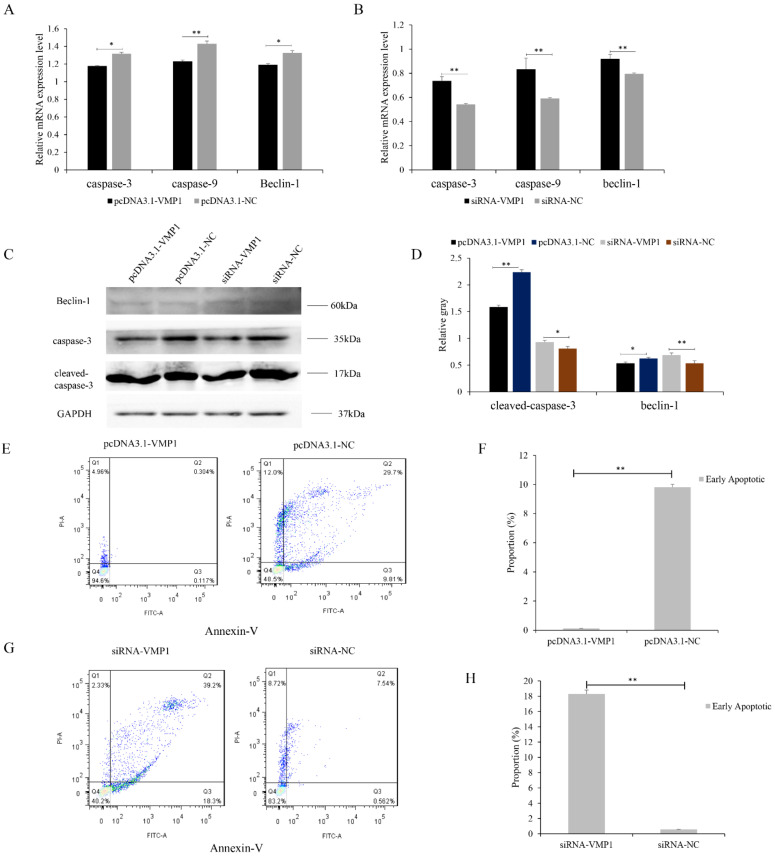
VMP1 regulates apoptosis of goat myoblasts: (**A**,**B**) Expression levels of apoptosis-related genes (caspase-3, caspase-9, and beclin-1) were quantified by RT-qPCR in myoblasts overexpressing and underexpressing VMP1. (**C**,**D**) Western blot analysis illustrates the levels of beclin-1 and caspase-3 cleavage in myoblasts overexpressing and underexpressing VMP1. GAPDH was used as the internal standard. (**E**) Apoptotic myoblasts were identified by flow cytometry. Cells were stained with annexin V–FITC/propidium iodide (PI). (**F**) Percentages of myoblasts in apoptosis. (**G**) Apoptotic myoblasts identified by flow cytometry. (**H**) Percentages of myoblasts in apoptosis. Replicates = 3. Data are presented as means ± SE; * *p* < 0.05 and ** *p* < 0.01. NC, negative control.

**Figure 5 cells-11-02227-f005:**
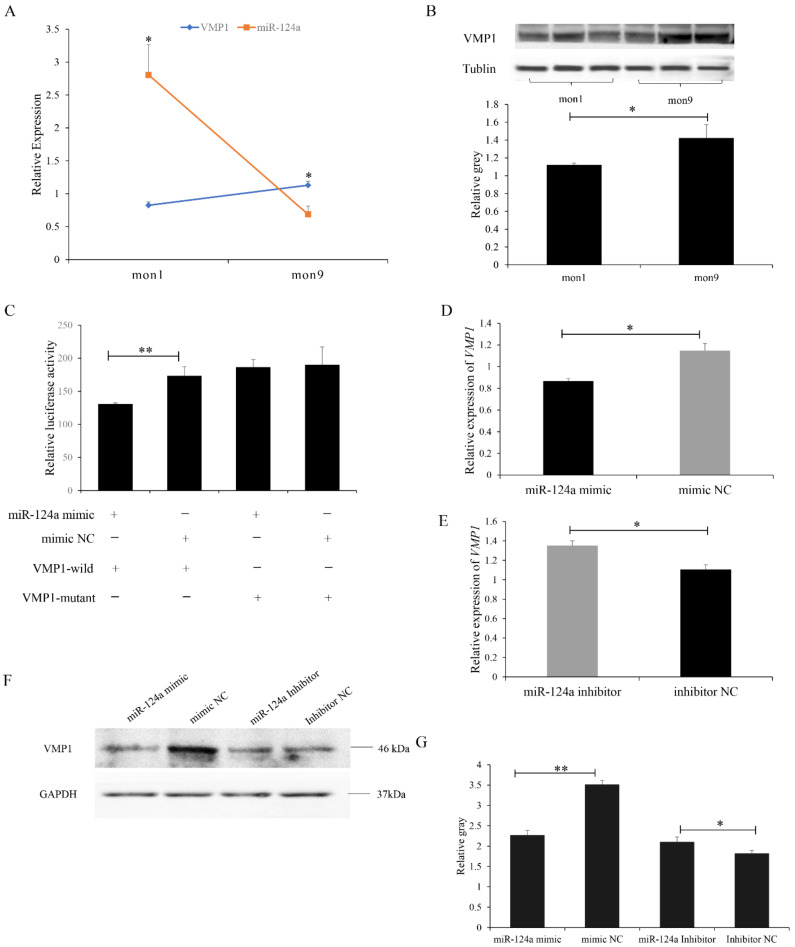
Expression of miR-124a and its regulation of VMP1 in goat LD tissue: (**A**) Expression levels of miR-124a and VMP1 in mon1 and mon9 goat LD tissues. Data are presented as means ± standard error (SE); *n* = 3. (**B**) Western blot of VMP1 levels in the LD tissues is consistent with mRNA expression. (**C**) Relative luciferase activity was assayed 48 h after 293T cells were co-transfected with VMP1-3′-UTR wild-type or mutant dual-luciferase vectors, together with the miR-124a mimic or mimic negative control (NC). (**D**,**E**) Expression of VMP1 in goat myoblasts was detected by RT-qPCR in cells overexpressing and underexpressing VMP1. (**F**,**G**) Western blot of VMP1 in goat myoblasts overexpressing and underexpressing miR-124a. Replications = 3. Data are presented as the mean ± SE; * *p* < 0.05 and ** *p* < 0.01.

**Figure 6 cells-11-02227-f006:**
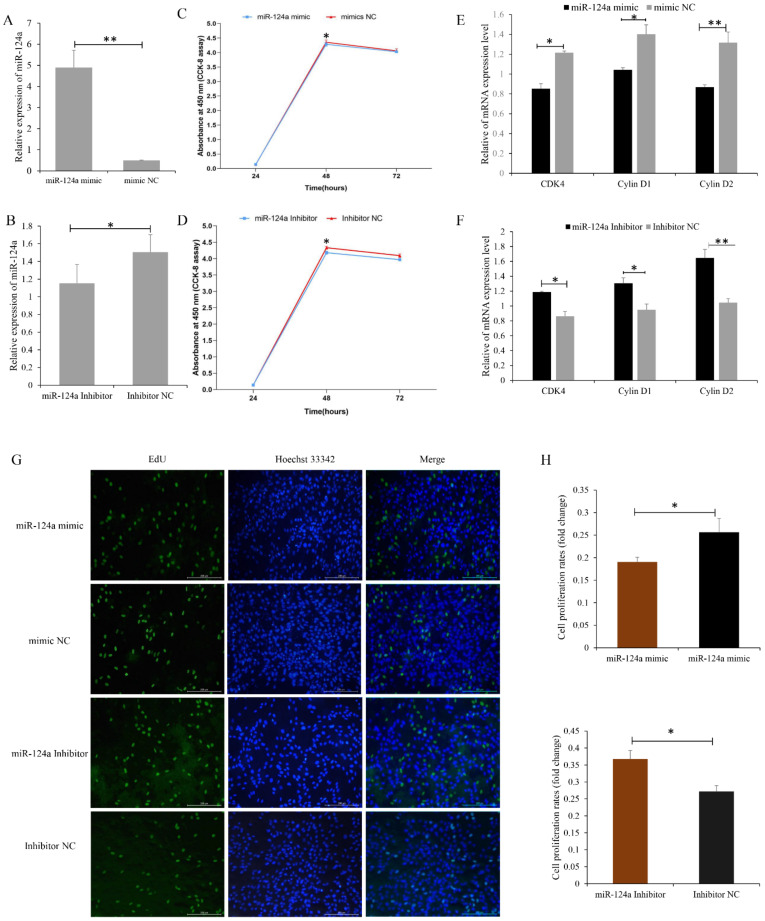
miR-124a regulates the proliferation of goat myoblasts: (**A**,**B**) RT-qPCR was used to determine miR-124a expression levels in myoblasts transfected with the miR-124a mimic and inhibitor. (**C**) Proliferation curve of myoblasts overexpressing miR-124a. (**D**) Proliferation curve of myoblasts underexpressing miR-124a. (**E**) Expression of the proliferation-related genes CDK4, cyclin D1, and cyclin D2 in cells overexpressing miR-124a. (**F**) Expression of proliferation-related genes in cells underexpressing miR-124a. (**G**) Myoblasts overexpressing and underexpressing miR-124a stained with EdU. (**H**) Fold change in the proliferation of myoblasts overexpressing and underexpressing miR-124a. Replicates = 3. Data are shown as the mean ± SE. * *p* < 0.05 and ** *p* < 0.01. NC, negative control.

**Figure 7 cells-11-02227-f007:**
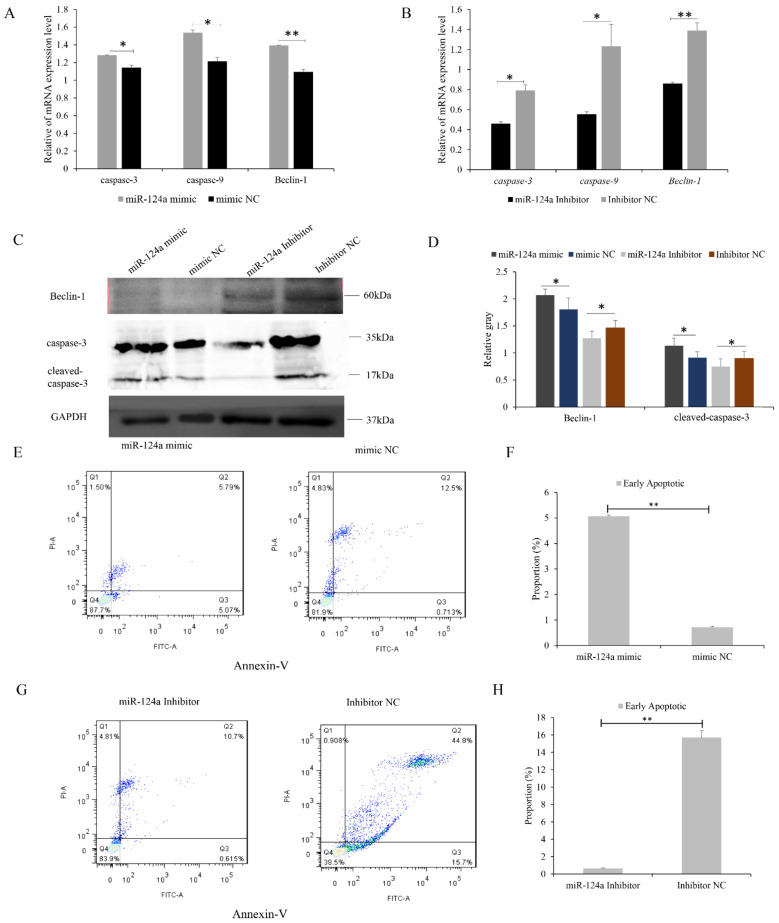
miR-124a regulates apoptosis of goat myoblasts: (**A**,**B**) Expression levels of cell-apoptosis-related genes (caspase-3, caspase-9, and beclin-1) in cells overexpressing and underexpressing miR-124a were quantified by RT-qPCR. (**C**,**D**) Western blot analysis revealed the expression of caspase-3 and beclin-1 after gain and loss of miR-124a. GAPDH was used as an internal reference. (**E**) Myoblasts overexpressing miR-124a were stained with annexin V–FITC/propidium iodide (PI), and then subjected to flow cytometry. (**F**) The percentage of miR-124a-overexpressing myoblasts in apoptosis. (**G**) Flow cytometry of myoblasts underexpressing miR-124a. (**H**) Percentage of miR-124a-underexpressing myoblasts in apoptosis. Replicates = 3. Data are presented as the mean ± SE; * *p* < 0.05 and ** *p* < 0.01. NC, negative control.

**Figure 8 cells-11-02227-f008:**
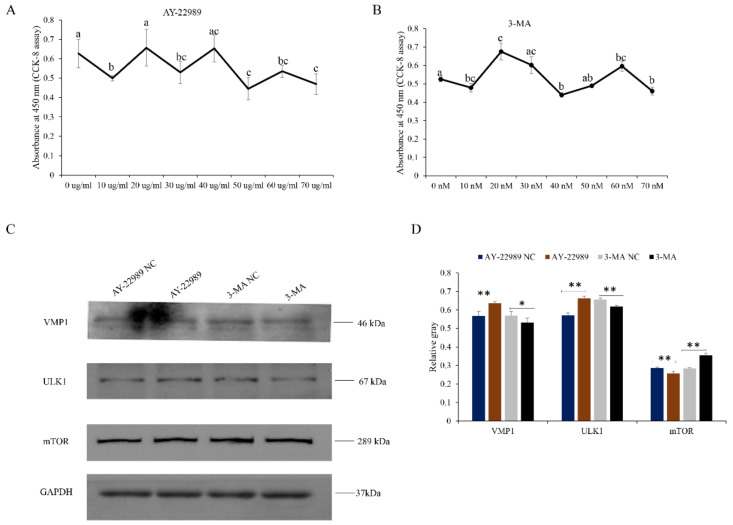
Phosphoinositide 3-kinase (PI3K/AKT/mTOR) pathway analysis: (**A**,**B**) CCK-8 assays were used to determine the optimal concentrations of the mTOR activator (AY-22989) and inhibitor (3-MA). (**C**,**D**) Levels of mTOR and downstream regulators (VMP1 and ULK1) in myoblasts were detected by Western blotting after treatment with AY-22989 or 3-MA. GAPDH was used as an internal control. Replicates = 3. Data are presented as the mean ± SE; * *p* < 0.05 and ** *p* < 0.01. The different letters represented the significant difference.

**Figure 9 cells-11-02227-f009:**
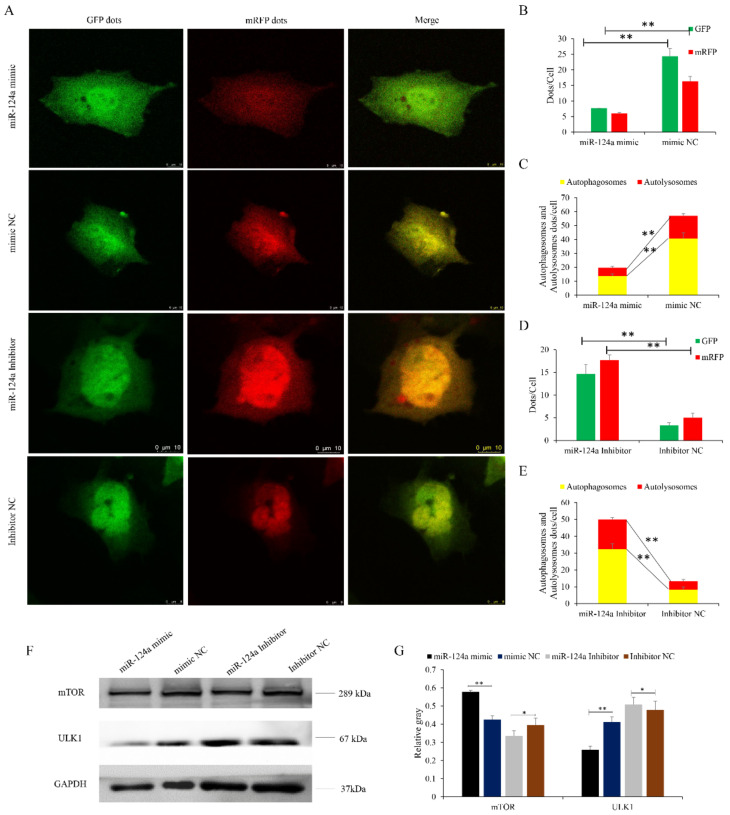
VMP1 promotes myoblast autophagy: (**A**) An adenovirus harboring tandem fluorescent mRFP-GFP-LC3 was used to evaluate the extent of autophagic flux after VMP1 overexpression or silencing. (**B**,**D**) Mean numbers of green and red puncta per cell; three cells were randomly selected from each field to be counted. (**C**,**E**) Mean numbers of autophagosomes and autolysosomes per cell. Autophagosomes contain both green and red puncta; in the merged images, the puncta appear yellow. Autolysosomes contain red puncta only. (**F**,**G**) Levels of ULK1/mTOR were detected after VMP1 overexpression or silencing. GAPDH was used as an internal standard. Replicates = 3. Data are presented as the mean ± SE; * *p* < 0.05 and ** *p* < 0.01.

**Figure 10 cells-11-02227-f010:**
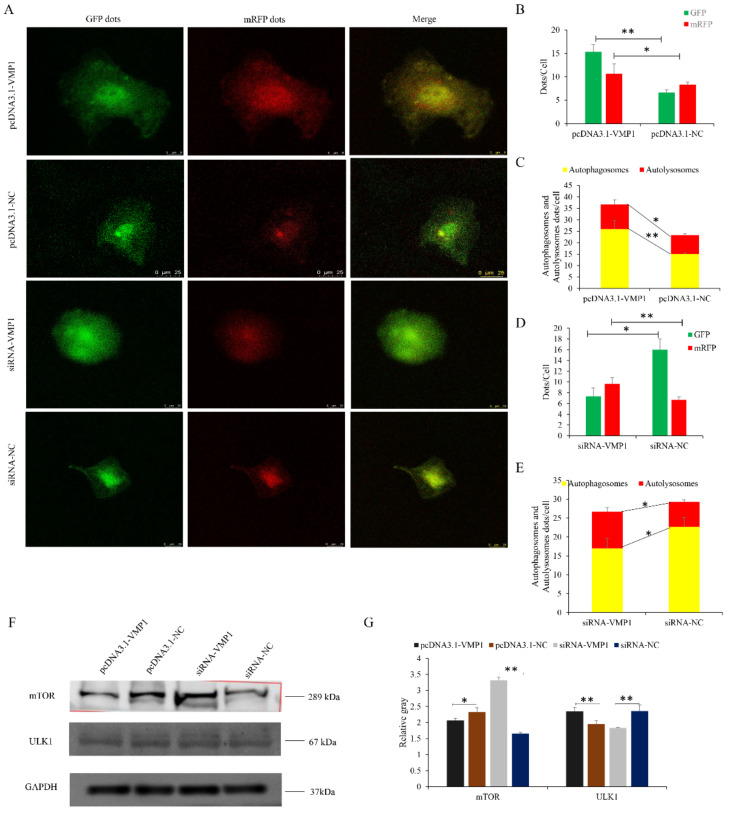
miR-124a regulates myoblast autophagy via the transient receptor potential of the VMP1/ULK1 pathway: (**A**) An adenovirus harboring tandem fluorescent mRFP-GFP-LC3 was used to evaluate the extent of autophagic flux after overexpression or inhibition of miR-124a. (**B**,**D**) Mean numbers of GFP and mRFP puncta per cell; three cells were randomly selected from each field to be counted. (**C**,**E**) Mean numbers of autophagosomes and autolysosomes per cell. Autophagosomes have green and red puncta, while in the merged images the puncta appear yellow. Autolysosomes have red puncta only. (**F**,**G**) Protein levels of ULK1/mTOR were detected after overexpression or inhibition of miR-124a. GAPDH was used as an internal standard. Replicates = 3. Data are presented as the mean ± SE; * *p* < 0.05 and ** *p* < 0.01.

**Figure 11 cells-11-02227-f011:**
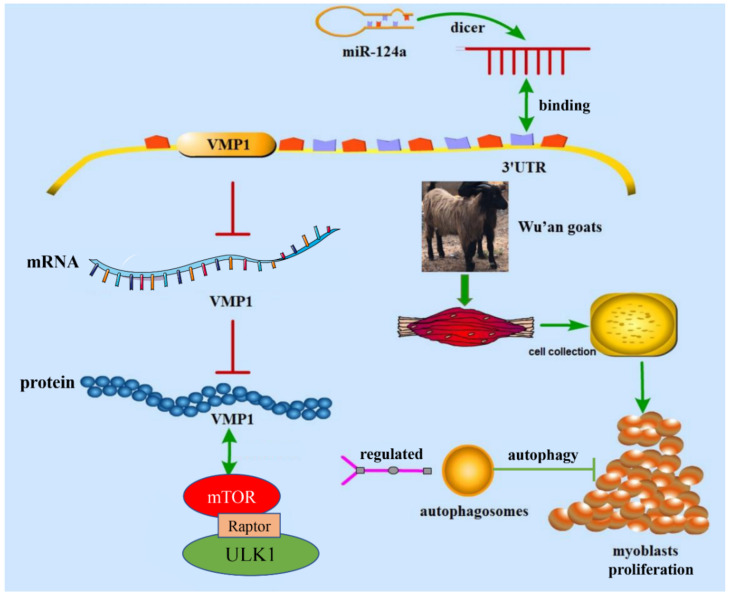
Flowchart of the effects of VMP1 regulated by miR-124a on the proliferation, autophagy, and apoptosis of goat myoblasts through the PI3K/AKT/mTOR signaling pathway.

## Data Availability

All datasets generated for this study can be found at Sequence Read Archive: PRJNA749569, and in the Appendix A.

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
