# Peer review of "VMP1 Regulated by chi-miR-124a Effects Goat Myoblast Proliferation, Autophagy, and Apoptosis through the PI3K/ULK1/mTOR Signaling Pathway"

_cells, 2022, doi:10.3390/cells11142227_

Round 1
Reviewer 1 Report
The article by Yufang Liu et al. details the muscle development in Wu’an goats, is associated with VMP1 expression mediated by miR-124a. The mechanisms underlying the skeletal muscle development during goat development are poorly understood and there are differences from in-other models. This paper is very comprehensive and describes studies performed in 1- and 9-month-old goats. The study shows that there is a paradoxical increase in VMP1 expression and miR-124a which leads to myoblast proliferation. The study analyzed using large set of RNA sequence data from goat samples and complemented with in-vitro cellular experiments. The article is well written and presented and the results are novel and should advance the field. However, the molecular mechanisms are little confusing and from the current form it is hard to draw conclusions. To address this the following issues should be addressed.
1. It is well established that VMP1 is a member of autophagic vesicle formation machinery and contacts between the ER and endosomes/endolysosomes are also reportedly regulated by interaction between VMP1 and SERCA. Based on this evidence it is very clear that VMP1 is important for autophagy in the cell. As authors mentioned autophagy is important for removal of damaged organelles withing the cell. So during early development both protein synthesis and break down are equally important and VMP1 plays one part of the proteostasis mechanism and removal of damaged organelles. The authors should consider addressing this in the intro or discussion.
2. The authors talk about systemic inflammation and cancer cachexia in both introduction and discussion and which I feel not necessary. In the early part of development myoblast proliferation is important and VMP1 is involved in the process of autophagy and apoptosis.
3. The authors stated that they have selected 10 DEGs and verified by RT-PCR. What is the basis of selecting these sets?
4. Both mTOR and ULK1 are regulated by phosphorylation and it would be nice if the authors can show phosphor mTOR/ULK1 by western blots.
Author Response
Thank you very much for evaluating our work! We have tried our best to improve the manuscript according to the reviewer’s further comments.
- It is well established that VMP1 is a member of autophagic vesicle formation machinery and contacts between the ER and endosomes/endolysosomes are also reportedly regulated by interaction between VMP1 and SERCA. Based on this evidence it is very clear that VMP1 is important for autophagy in the cell. As authors mentioned autophagy is important for removal of damaged organelles withing the cell. So during early development both protein synthesis and break down are equally important and VMP1 plays one part of the proteostasis mechanism and removal of damaged organelles. The authors should consider addressing this in the intro or discussion.
Response: Thank you for your suggestion! We have addressed the function of VMP1 in the early development both protein synthesis and break down (line 479-487).
- The authors talk about systemic inflammation and cancer cachexia in both introduction and discussion and which I feel not necessary. In the early part of development myoblast proliferation is important and VMP1 is involved in the process of autophagy and apoptosis.
Response: Thank you very much! We have removed this part in the new version.
- The authors stated that they have selected 10 DEGs and verified by RT-PCR. What is the basis of selecting these sets?
Response: Thank you very much! To validate the accuracy of RNA-seq data, ten DEGs was randomly selected to verify by RT-qPCR.
- Both mTOR and ULK1 are regulated by phosphorylation and it would be nice if the authors can show phosphor mTOR/ULK1 by western blots.
Response: Thank you for your suggestion! It’s very important and useful! In this study, we preliminary explored that VMP1 was affected by miR-124a and mTOR signalling pathway to regulate myoblast autophagy and proliferation. In the future, we will be to investigate the relationship between VMP1 and mTOR signalling pathway in depth, and will design the experiments related to phosphorylation. Thank you very much!
Reviewer 2 Report
Cells-1743685
VMP1 regulated by chi-miR-124a inhibits goat myoblast apoptosis through the PI3K/ULK1/mTOR signaling pathway promotes proliferation and autophagy
Summary: The authors utilize RNA-seq of total RNA isolated from the longissimus dorsi from a specific species of goat native to China. The typical determination of differential abundance, ontology and pathway analysis was performed. Selected genes were chosen for conformation of RNA-seq data; however, it was interesting that the gene, VMP1 was not chosen to be examined via RT-qPCR. The authors tie in miR-124a and the PI#K/ULK1/mTOR signaling pathway to cell proliferation and autophagy.
Critique:
There are several concerns the authors most address to be acceptable for publication.
The materials and methods section needs a complete rewrite. More information on how RT-qPCR of miRNA was conducted. Primers, primer sequences and efficiencies or all primers used in PCR need to be included, especially if using the 2-delta delta Ct method.
The RIN (mean) and 260/280 information needs to be included in the manuscript to evaluate quality of the starting material for sequencing and RT-PCR. Further, was any validation (bestkeeper analysis conducted) for GAPDH and/or U6. Without this analysis the RT-qPCR data is meaningless.
Verification of differential abundance of RNA-seq data.
It is very concerning that the VMP1 gene was not included in verification using RT-qPCR. Please include this experiment and data in the paper. Further, the authors need to a description of the samples. Were the samples the same total RNA from the 6 goats used in RNA-seq. If so, additional information as to validate the relative abundance of VMP1 and the other genes need to be conducted in tissues from other animals and probably in numbers of ~5 per age group. Further where the goats utilized all from the same billy? Multiple billies? What was the sex of the 6 goats? More information on the goats is required.
Figures 3 and 4.
The experimental design in figures 3 and 4 are flawed. A pDNA3.1-VMP1 vector with WT and mutated 3’ UTR transfected in the presence or absence of siRNA-VMP1 is required to show a direct association of miR-124a with VMP1 and linking them both the proliferation and autophagy being examined. Without that the only conclusion that can be made as they both impacted specific abundance of some of the same genes. Again, where is the western blotting experiments showing a direct association of miR-124 a with VMP1 protein abundance. This may have been performed as described in figure 5; however, the vector labeling does not match the materials and methods section, nor was the impact of the minimal alteration of VMP1 evaluated on the subsequent signaling pathway.
On a more constructive note; the authors do give a nice diagram of how the proposed pathway is affected by VMP1 and miR-124a. While their data supports that diagram, further experiments with the reagents they have already generated need to be conducted to verify that diagram.
Other:
1. The paper could use a rewrite and correction of English.
2. Throughout the paper, the authors refer to expression and differential expression. Expression levels or differential expression indicates that promoter activity and the synthesis of RNA measured. Nowhere in the paper was this performed. The authors measured abundance and differential abundance of genes. Abundance refers to the quantification (in this case semi-quantification) of genes chosen. This abundance can be impacted by expression levels, RNA stability and/or RNA degradation. Please correct throughout the paper.
Recommendations: Reject. Resubmit after requested information and experiments are conducted.
Author Response
Thank you very much for evaluating our work! We have tried our best to improve the manuscript according to the reviewer’s further comments.
Critique:
There are several concerns the authors most address to be acceptable for publication.
The materials and methods section needs a complete rewrite. More information on how RT-qPCR of miRNA was conducted. Primers, primer sequences and efficiencies or all primers used in PCR need to be included, especially if using the 2-delta delta Ct method.
Response: Thank you very much! To avoid duplication of content, we have added references of the 2-△△Ct methods (line 142). The information of primers, primer sequences and efficiencies or all primers used in PCR were shown in supplementary material Table S1.
The RIN (mean) and 260/280 information needs to be included in the manuscript to evaluate quality of the starting material for sequencing and RT-PCR. Further, was any validation (bestkeeper analysis conducted) for GAPDH and/or U6. Without this analysis the RT-qPCR data is meaningless.
Response: Thank you very much! The RIN (mean) and 260/280 information were added in the new version (line 115-118). The housekeeping genes GAPDH and U6 were validated by bestkeeper and can be used as reference gene for subsequent experiments.
Verification of differential abundance of RNA-seq data.
It is very concerning that the VMP1 gene was not included in verification using RT-qPCR. Please include this experiment and data in the paper. Further, the authors need to a description of the samples. Were the samples the same total RNA from the 6 goats used in RNA-seq. If so, additional information as to validate the relative abundance of VMP1 and the other genes need to be conducted in tissues from other animals and probably in numbers of ~5 per age group. Further where the goats utilized all from the same billy? Multiple billies? What was the sex of the 6 goats? More information on the goats is required.
Response: Thank you very much! To compare the expression trends of VMP1 and miR-124a, we have placed the VMP1 expression results in Figure 5A. In this study, five samples per group were used to the RT-qPCR validation, including 3 individuals for the RNA-seq. We have revised in the new version (137-138).
Figures 3 and 4.
The experimental design in figures 3 and 4 are flawed. A pDNA3.1-VMP1 vector with WT and mutated 3’ UTR transfected in the presence or absence of siRNA-VMP1 is required to show a direct association of miR-124a with VMP1 and linking them both the proliferation and autophagy being examined. Without that the only conclusion that can be made as they both impacted specific abundance of some of the same genes. Again, where is the western blotting experiments showing a direct association of miR-124 a with VMP1 protein abundance. This may have been performed as described in figure 5; however, the vector labeling does not match the materials and methods section, nor was the impact of the minimal alteration of VMP1 evaluated on the subsequent signaling pathway.
On a more constructive note; the authors do give a nice diagram of how the proposed pathway is affected by VMP1 and miR-124a. While their data supports that diagram, further experiments with the reagents they have already generated need to be conducted to verify that diagram.
Response: Thank you very much! We are very sorry that we could not understand the reviewers' comments accurately. However, the Figure 3 and Figure 4 illustrate that VMP1 expression was associated with myoblast proliferation and apoptosis and does not address the regulation of VMP1 by miR-124a. The miR-124a regulation of VMP1 was shown in Figure 5.
Other:
- The paper could use a rewrite and correction of English.
Response: Thank you very much! The manuscript has been revised by the English native speaker.
- Throughout the paper, the authors refer to expression and differential expression. Expression levels or differential expression indicates that promoter activity and the synthesis of RNA measured. Nowhere in the paper was this performed. The authors measured abundance and differential abundance of genes. Abundance refers to the quantification (in this case semi-quantification) of genes chosen. This abundance can be impacted by expression levels, RNA stability and/or RNA degradation. Please correct throughout the paper.
Response: Thank you very much! We have revised in the new version.
Recommendations: Reject. Resubmit after requested information and experiments are conducted.
Reviewer 3 Report
In this paper, through transcriptome sequencing, cell culture, cell transfection and other experiments, we studied the pathway of chi-mir-124a regulating good myoblast apoptosis through the PI3K/ULK1/mTOR signaling pathway. The overall experimental design is complete and eye-catching, but there are still some problems:
1. The title has a long number of words, which is not conducive to reading. It is recommended to simplify it.
2. Compared with the picture in Figure 2 e,the thickness of the X-axis and Y-axis of the histogram in Figure 1, 3, 4, 5, 6, 7, 8, 9 and 10 is too light to be read. It is recommended to unify them. At the same time, these pictures were completed by Excel and GraphPad prism respectively, which need to be listed in the method.
3. In Fig. 2B, it can be seen that the expression trends of most genes in the groups of mon9 -2, and mon9-3 are more similar with mon1 rather than with mon9-1. Please provide the PCA principal component analysis of the six samples. If the sample clustering shows that the differences among the three sample groups of mon9 are larger than the differences between mon1 and mon9, please give a reasonable explanation. In addition, the current heat map cannot show the label on the right clearly. It is suggested to label the representative genes in each part of the genes separately, such as the genes highly expressed in mon9 and low expressed in mon1.
4. Caspase-3 was detected in Figure 4C, however the marker of apoptotic cells is cleaved-caspase-3, not Caspase-3. The detection results of cleaved-caspase-3 need to be supplemented.
5. In Fig. 4 and Fig. 7, the vertical and horizontal coordinates corresponding to the "Crossgate" of the results of apoptosis detection need to be consistent in the same comparison group, for example, pcdna3.1-vmp1 and pcdna3.1-NC in Fig. 4 E. Non-unified treatment may reduce the reliability of the results, so the results need to be reanalyzed. Meanwhile,these results (Fig. 4 F, h and Fig. 7 F, H) need to be marked with the significance of differences as in Fig. 9 C and E.
Author Response
Thank you very much for evaluating our work! We have tried our best to improve the manuscript according to the reviewer’s further comments.
- The title has a long number of words, which is not conducive to reading. It is recommended to simplify it.
Response: Thank you very much! We have revised in the new version.
- Compared with the picture in Figure 2 e,the thickness of the X-axis and Y-axis of the histogram in Figure 1, 3, 4, 5, 6, 7, 8, 9 and 10 is too light to be read. It is recommended to unify them. At the same time, these pictures were completed by Excel and GraphPad prism respectively, which need to be listed in the method.
Response: Thank you very much! Because there are so many images in a figure, the font sizes on the X- and Y-axes of the histogram are relatively small and we have added tick marks to improve readability. The software has been listed in the new version (line 231).
- In Fig. 2B, it can be seen that the expression trends of most genes in the groups of mon9 -2, and mon9-3 are more similar with mon1 rather than with mon9-1. Please provide the PCA principal component analysis of the six samples. If the sample clustering shows that the differences among the three sample groups of mon9 are larger than the differences between mon1 and mon9, please give a reasonable explanation. In addition, the current heat map cannot show the label on the right clearly. It is suggested to label the representative genes in each part of the genes separately, such as the genes highly expressed in mon9 and low expressed in mon1.
Response: Thank you very much! We have revised Figure 2B, and added the genes in the new version. Due to the image from the PCA analysis was too large to be placed in Figure 2, we have placed it in Figure S1 as supplementary material. The PCA results showed that the mon1 group and mon9 group clustered together, respectively. But there was more variation between individuals in the mon9 group, and in fact the DEGs obtained in this way were more representative of the true differences.
- Caspase-3 was detected in Figure 4C, however the marker of apoptotic cells is cleaved-caspase-3, not Caspase-3. The detection results of cleaved-caspase-3 need to be supplemented.
Response: Thank you very much! We have added the western blot assay of cleave-caspase-3 in the new version.
- In Fig. 4 and Fig. 7, the vertical and horizontal coordinates corresponding to the "Crossgate" of the results of apoptosis detection need to be consistent in the same comparison group, for example, pcdna3.1-vmp1 and pcdna3.1-NC in Fig. 4 E. Non-unified treatment may reduce the reliability of the results, so the results need to be reanalyzed. Meanwhile, these results (Fig. 4 F, h and Fig. 7 F, H) need to be marked with the significance of differences as in Fig. 9 C and E.
Response: Thank you very much! We have revised in the new version.
Round 2
Reviewer 2 Report
The authors have addressed most points of the previous review. The article is acceptable.
Author Response
Thank you very much!
Reviewer 3 Report
The contents were improved obviously, however these's still some problems:
1. The current title is reasonable.
2. The picture is much better now than before.
3. Sorry, I didn't find Figure S1 in the supplementary materials.
4. It is recommended to detect caspase-3 and cleared caspase-3 proteins on the same membrane, and optimize the application of antibodies.
5. It is a good choice to seek the help of people who are proficient in the analysis of flow cytometry results.
Author Response
The contents were improved obviously, however these's still some problems:
- The current title is reasonable.
Answer: Thank you very much!
- The picture is much better now than before.
Answer: Thank you very much!
- Sorry, I didn't find Figure S1 in the supplementary materials.
Answer: Thank you very much! We have re-upload the Figure S1 in the supplementary materials, please check it out!
- It is recommended to detect caspase-3 and cleared caspase-3 proteins on the same membrane, and optimize the application of antibodies.
Answer: Thank you very much! We have replaced the WB result of caspase-3 and cleared caspase-3 proteins in the new version.
- It is a good choice to seek the help of people who are proficient in the analysis of flow cytometry results.
Answer: Thank you very much!